# Influence of Aliphatic Chain Length on Structural, Thermal and Electrochemical Properties of *n*-alkylene Benzyl Alcohols: A Study of the Odd–Even Effect

**DOI:** 10.3390/molecules27123781

**Published:** 2022-06-12

**Authors:** Tomislav Balić, Marija Paurević, Marta Počkaj, Martina Medvidović-Kosanović, Dominik Goman, Aleksandar Széchenyi, Zsolt Preisz, Sándor Kunsági-Máté

**Affiliations:** 1Department of Chemistry, Josip Juraj Strossmayer University of Osijek, Cara Hadrijana 8/A, 31000 Osijek, Croatia; tombalic@kemija.unios.hr (T.B.); mstivo@kemija.unios.hr (M.P.); dgoman@kemija.unios.hr (D.G.); szealex@kemija.unios.hr (A.S.); 2Faculty of Chemistry and Chemical Technology, University of Ljubljana, Večna pot 113, 1000 Ljubljana, Slovenia; marta.pockaj@fkkt.uni-lj.si; 3Faculty of Pharmacy, Institute of Organic and Medicinal Chemistry, University of Pécs, Szigeti 12, 7624 Pécs, Hungary; preisz.zsolt@gmail.com (Z.P.); kunsagi-mate.sandor@gytk.pte.hu (S.K.-M.); 4Department of Physical Chemistry and Materials Science, Faculty of Sciences, University of Pécs, Ifjúság 6, 7624 Pécs, Hungary; 5János Szentágothai Research Center, University of Pécs, Ifjúság 20, 7624 Pécs, Hungary

**Keywords:** aromatic alcohols, aliphatic chain, odd–even effect, melting point deviation, crystal structures

## Abstract

The century-old, well-known odd–even effect phenomenon is still a very attractive and intriguing topic in supramolecular and nano-scale organic chemistry. As a part of our continuous efforts in the study of supramolecular chemistry, we have prepared three novel aromatic alcohols (1,2-bis[2-(hydroxymethyl)phenoxy]butylene (**Do4OH**), 1,2-bis[2-(hydroxymethyl)phenoxy]pentylene (**Do5OH**) and 1,2-bis[2-(hydroxymethyl)phenoxy]hexylene (**Do6OH**)) and determined their crystal and molecular structures by single-crystal X-ray diffraction. In all compounds, two benzyl alcohol groups are linked by an aliphatic chain of different lengths (CH_2_)_n_; *n* = **4**, **5** and **6**. The major differences in the molecular structures were found in the overall planarity of the molecules and the conformation of the aliphatic chain. Molecules with an even number of CH_2_ groups tend to be planar with an all-*trans* conformation of the aliphatic chain, while the odd-numbered molecule is non-planar, with partial *gauche* conformation. A direct consequence of these structural differences is visible in the melting points—odd-numbered compounds of a particular series display systematically lower melting points. Crystal and molecular structures were additionally studied by the theoretical calculations and the melting points were correlated with packing density and the number of CH_2_ groups. The results have shown that the generally accepted rule, higher density = higher stability = higher melting point, could not be applied to these compounds. It was found that the denser packaging causes an increase in the percentage of repulsive H‧‧‧H interactions, thereby reducing the stability of the crystal, and consequently, the melting points. Another interesting consequence of different molecular structures is their electrochemical and antioxidative properties—a non-planar structure displays the highest oxidation peak of hydroxyl groups and moderate antioxidant activity.

## 1. Introduction

Aromatic dialcohols containing a different number of methylene groups in the aliphatic chain are well known as a precursor in the synthesis of a vast number of azaoxa and thiaoxa macrocycles [1]. It is also important to emphasize that the change in chain length can be used to manipulate the size of the binding space in the macrocyclic system, and thus, to modify the binding affinity towards different chemical species. As previously described by our group in the case of dialdehydes, these subtle changes in the chain length can lead to the different extraction properties of macrocycles [2,3]. Further on, such changes result in diverse conformations of macrocycles, with an important impact on the dimensionality and supramolecular assembly of coordination compounds [4].

It was recognized almost 120 years ago [5] that *n*-alkyl carboxylic acids do not undergo a constant increase in melting point depending on the length of the aliphatic chain. Instead of such behavior, compounds with an even number of carbons have higher melting points in comparison to molecules with an odd number of carbon atoms of the same series. Such behavior is known as the odd–even effect [6]. The origin of this effect was attributed to the optimal intermolecular and interlayer interactions between molecules in the solid-state [7] and usually correlated to packing densities. This effect was also observed in a series of *n*-alkanes [8], *α*,*ω*-alkendiamines [8], *α*,*ω*-alkendiols [8] and *α*,*ω*-alkendithiols [9]. In the case of *n*-alkanes, the melting point is directly correlated to the packing density—the odd-numbered *n*-alkanes having lower density, while in the case of *α*, *ω*-dicarboxylic acids, the effect is just the opposite. This contradiction was explained by the intermolecular arrangement of molecules in crystalline state that maximize van der Waals interactions between adjacent molecules and layers of molecules [8]. Of particular interest to our investigation is the study of the odd–even effect in *α*,*ω*-alkendiols conducted by Thalladi et al. [8]. This systematic investigation has shown three important conclusions: (*i*) the OH group acts both as a hydrogen bond donor and acceptor and represents the primary mode of intermolecular connection; (*ii*) the conformation of the aliphatic chain in even-membered molecules is *all*-trans, leading to offset packing of aliphatic chain and more dense packing with favorable hydrophobic interactions; (*iii*) odd-membered diols display a partial *gauche* conformation of the aliphatic chain, *leading* towards the formation of 3D supramolecular networks with less-effective hydrophobic interactions. Certainly, the most interesting consequence of the odd–even effect was found in *n*-alkenthiolates. These compounds were used as self-assembled monolayers (SAMs) in Ag/GaOx/EGaIn electrode systems. It was demonstrated in this work that odd-numbered molecules impede the charge transfer process in the investigated system [10].

In order to further contribute to the clarification of the odd–even effect phenomenon, we have prepared and structurally characterized three novel *n*-alkylene aromatic alcohols. The compounds were characterized by means of elemental analysis, IR and NMR spectroscopy. The molecular and crystal structures were determined by the single-crystal X-ray diffraction method. The structure–melting point–electrochemical property correlation was conducted by comparison with previously published chain analogs ((CH_2_)_n_; n = 1, 2, 3) using theoretical calculations (Hirshfeld surface analysis and interaction energy calculations), thermal analysis and cyclic voltammetry measurements.

## 2. Materials and Methods

All chemicals were procured from commercial sources and were of reagent grade. IR spectra were recorded on a Shimadzu FTIR 8400S spectrophotometer using a DRS 8000 attachment, in the 4000–400 cm^−1^ region. Thermogravimetric analyses were performed using a simultaneous Mettler Toledo TGA/DSC 1. The samples were heated in a nitrogen atmosphere (200 mL min^−1^) in 100 μL aluminum pans up to 500 °C at a rate of 5 °C min^−1^. Before measurements, a blank curve measurement under the same experimental conditions was run, and the blank curve was subtracted. The data collection and analyses were performed using the program package STAR^e^ Software 10.0 [11]. A DSC analysis of **Do6OH** polymorphic forms **I** and **II** was performed using a Setaram MicroSC from 25 °C to 150 °C at a rate of 1.2 °C min^−1^ in Hastelloy C crucibles. The ^1^H NMR spectra were recorded on a Bruker Avance III HD NMR 500 MHz instrument at 500 MHz, using deuterated DMSO as solvent. All NMR experiments were performed at 298 K. All coupling constants are reported in hertz (Hz). Chemical shifts are reported in ppm and are referenced to residual solvent peak DMSO (^1^H 2.5 ppm). Stock solutions of investigated compounds (**Do4OH**, **Do5OH** and **Do6OH**) were prepared in DMF. Before each measurement, an appropriate amount of each compound was diluted in 0.1 M KOH. Electrochemical experiments were performed on a PalmSens potentiostat/galvanostat (PalmSens BV, Utrecht, The Netherlands) driven by PSTrace 4.2 software. A conventional three-electrode cell was used with a gold electrode as a working electrode, Ag/AgCl as a reference electrode and a platinum wire as a counter electrode. The gold working electrode was polished with polishing α-Al_2_O_3_ (0.05 µm, ALS, Japan) before each measurement. The cyclic voltammetry scan rate varied from 50 mV s^−1^ to 400 mV s^–1^. Fresh DPPH (c(2,2-diphenyl-1-picrylhydrazyl) = 3 × 10^−4^ mol dm^−3^) solution was prepared in methanol. The reaction of radical scavenging was carried out by mixing the 100 µL of the investigated sample with 500 µL of the DPPH solution, and samples were kept in the dark and covered with foil for 15 min. The UV-Vis measurements were performed using a Shimadzu UV-2600 Spectrophotometer at λ_max_ = 518 nm. The antioxidant activity of samples was evaluated by changes in colors, from dark purple to pale yellow. The measurements were performed in triplicate and expressed as % scavenging activity (% DPPH).

### 2.1. X-ray Crystallography

The single-crystal X-ray diffraction data were collected at 150 K on an Oxford Diffraction SuperNova CCD diffractometer equipped with an Atlas detector and with graphite-monochromated Mo-*K*_α_ radiation (λ = 0.71073 Å) using *ω*–scans. The data reduction was performed using CrysAlis Pro [12]. Structures were solved by ShelXT [13] using intrinsic phasing and refined by a full-matrix least-squares procedure based on *F*^2^ with SHELXL-2018/3 [14]. All non-hydrogen atoms were refined anisotropically, and hydrogen atoms in the structures were initially located in difference Fourier maps; those attached to C-atoms were afterward placed in calculated positions and refined using the riding model, while those residing on oxygen atoms were refined freely. Geometrical calculations and Newman projections were made using PLATON [15,16], and structure drawings were made with the MERCURY [17] program. The crystallographic data are summarized in Appendix A.

### 2.2. Interaction Energy Calculation and Hirshfeld Surface Analysis

CrystalExplorer 17.5 was used to analyze Hirshfeld surfaces and evaluate the interaction energies and Hirshfeld surface analysis, using CIF files as input [18]. The energy calculations were performed with a B3LYP/6-31+G(d,p) computational model [18]. Scale factors used for calculations were set according to the used level of theory. Crystal lattice energies were calculated as one-half of the product of the two columns labeled *N* and *E*_tot_, where *N* is the number of molecule pairs in the cluster with that particular interaction energy using data obtained from interaction energy calculation (Appendix A). In these calculations, a symmetry-independent molecule was selected, and energy interactions with adjacent molecules in the 3.8 Å cluster radius were calculated. For compounds containing more than 1 molecule in the asymmetric unit, calculations were made for both conformer molecules independently. Energies with the same value of distance (R) were counted only once. Crystal lattice energies were also calculated using the PixelC [19] program implemented in the Oscail software package. The obtained CIF files were imported in the Oscail program and PixelC was run using the Oscail interface (Run Job) to convert files into appropriate format. Using PixelC, the centroids and centroid contacts were calculated with a maximum contact distance of 20 Å. Thus, prepared files were opened in PixelC and lattice energy was calculated using Orca [20] with MultiWFN [21] and with a B3LYP computational model [18]. The computations were performed following the previously described procedure [22]. The Hirshfeld surfaces were plotted over normalized contact distances (*d*_norm_) with standard color settings: regions highlighted in red represent short contacts while longer contacts are shown in blue. White areas represent contacts at the distance equal to the sum of the van der Waals radii.

Additionally, the full and resolved 2D fingerprint plots that show distances from each point on the Hirshfeld surface to the nearest atom inside (*d*_i_) and outside (*d*_e_) of it were calculated for all four structures.

### 2.3. Synthesis

Compounds were prepared by the reduction of appropriate aldehyde groups with NaBH_4_, following a previously reported method [23], according to Appendix A. Dialdehyde precursors **1**, **2**, **3** were synthesized by previously reported methods [3,24,25]. Details regarding the synthesis procedure can be found in ESI. The recrystallization of the **Do6OH** from two different solvent systems (DMF/H_2_O and DMSO/H_2_O) resulted in the formation of two polymorphs (Forms **I** and **II**, respectively).

## 3. Results

### 3.1. Description of Crystal Structures

Investigated compounds crystallize in the orthorhombic (**Do5OH** and **Do6OH II**) and monoclinic (**Do4OH** and **Do6OH I**) crystal systems. The molecular structures of compounds are shown in Figure 1 and Figure 2. The selected bond lengths and angles are presented in Appendix A. In all four compounds two benzyl alcohol units are connected by an aliphatic chain of different lengths, (CH_2_)_n_, with n = 4, 5 and 6. The primary interest of this work is to investigate the impact of aliphatic chain length on molecular and crystal structure. In all four crystal structures, the benzyl alcohol moieties are oriented opposite to each other (*anti*-orientation). The influence of aliphatic chain length on the molecular structure of prepared compounds (and consequently, on the crystal packing arrangement and thermal properties) can be determined by using two parameters: the deviation of benzene rings from planarity and the conformation of the aliphatic chain.

The asymmetric unit of **Do4OH** is composed of two symmetrically independent half molecules. In each molecule, the crystallographic center of inversion lies between two central atoms of the aliphatic chain (C9–C9i and C18–C18i). The aliphatic chain is in an all-*trans* conformation (Table 1 and Appendix A) and molecules are essentially planar, with negligible deviations of atoms from the plane calculated through the molecule. Unlike **Do4OH**, the compound **Do5OH** is a non-planar molecule with a dihedral angle between the benzene rings of 84.2(1)°. The torsion angle O2–C8––C9–C10 (−62.67(11)°) indicates a partial *gauche* conformation of the aliphatic chain (*trans*–*gauche*–*trans*–*gauche*–*trans* conformation of aliphatic chain starting from C7 atom). It is interesting to note that the OH∙∙∙OH distance of alcohol groups is significantly shorter in compounds with 5 CH_2_ groups (11.133(2) Å) in comparison to **Do4OH** (13.754(2) Å).

The recrystallization of the **Do6OH** resulted in the formation of two polymorphs (Forms **I** and **II**). Both forms were obtained as transparent needle-like crystals. Form **I** crystallized in the monoclinic crystal system (*P*2_1_/*c*) and **II** in the orthorhombic crystal system (*Pbca*). The asymmetric unit of form **I** is composed of two molecules (conformer **A** and **B**—Figure 2a). As in **Do4OH**, in each molecule, the crystallographic center of inversion lies between two central atoms of the aliphatic chain (C10–C10i and C20–C20i). The aliphatic chain is in an all-*trans* conformation (Appendix A) and molecules are essentially planar. However, in this case (form **I**), there is another difference in the conformation of molecules—in conformer **A**, alcohol groups are oriented away from the plane of the molecule, while in **B**, alcohol groups reside in the plane of the molecule (structure overlay can be found in ESI—Appendix A). These differences are also observable (Appendix A) in values of torsion angles (C7–C2–C1–O1 and C17–C12–C11–O3). Considering the molecular structure of form **II**, the largest difference is in the aliphatic chain conformation in comparison to form **I**. In form **II**, the aliphatic chain is in a partial *gauche* conformation (*trans*–*gauche*–*trans*–*trans*–*trans*–*gauche*–*trans* conformation of aliphatic chain starting from C7 atom). Due to this conformation, the molecule has the shape of a ladder (structure overlay with form **I**, conformer **A** can be found in ESI—Appendix A).

As expected, in all crystal structures, molecules are primarily connected by strong O−H∙∙∙O hydrogen bonds through terminal alcohol groups. Although rather similar compounds, there are some major differences in their crystal packing arrangements. In **Do4OH**, molecules are connected via O−H∙∙∙O hydrogen bonds (Figure 3). Additional stabilization of the crystal structure is achieved by a series of weak C−H∙∙∙O van der Waals interactions and C−H∙∙∙ π interactions of adjacent symmetrically independent molecules (Table 2).

In contrast to the above-described structure, **Do5OH** molecules are connected by strong O−H∙∙∙O hydrogen bond interactions into infinite chains approximately along an *a*-axis. These interactions form a staircase-like motif (Figure 4). Two adjacent chains (staircases) are connected by O−H∙∙∙O hydrogen bonds along the *b*-axis. The final arrangement of molecules in the crystal structure is achieved by a series of weak C−H∙∙∙O van der Waals interactions along the *c*-axis.

Differences in the molecular structure of **Do6OH** polymorphs caused variations in packing arrangements and intermolecular interactions. In the structure of form **I**, conformer molecules (**A** and **B**) are linked via strong O−H∙∙∙O hydrogen bond interactions (Table 2). In both conformers, terminal oxygen atoms (O1, O1i, O3 and O3i) form two hydrogen bonds (bifurcated) with the adjacent conformer molecules’ interactions forming a supramolecular staircase motif (Figure 5a).

A rather similar packing arrangement is observed in the crystal structure of form **II**—bifurcated terminal oxygen atoms (O1 and O1i) connected by strong O−H∙∙∙O hydrogen bonds. Such interactions lead to the formation of 2D sheets, connected by weak C−H∙∙∙O van der Waals interactions along the *b*-axis (Figure 5b).

To further correlate the properties of the investigated compounds, a Cambridge Structural Database (CSD) [29] search was performed for compounds containing 2-(hydroxyformyl)phenoxy units linked by a different number of methylene groups in the aliphatic chain. The search resulted in three relevant hits, and the query results are presented in Table 1. Some common features of the compounds can be summarized: (i) pendant alcohol groups are *anti-oriented* in all compounds except in SAZCIF, resulting in an unusually short OH‧‧‧OH distance (6.446(4)) in this compound; (ii) molecules with an even number of methylene groups in the aliphatic chain are planar (except SAZCIF), while molecules with an odd number display significant deviation from planarity (see dihedral angles in Table 1); (iii) the conformation of the aliphatic chain is all-*trans* for molecules with an even number of methylene groups (except in SAZCIF), while in odd-numbered molecules, a different pattern of chain conformation is observed. The impact of the molecular and crystal structures of compounds on their thermal properties is discussed in the thermal analysis section (vide infra).

### 3.2. Hirshfeld Surface (HS) Analysis and Interaction Energy Calculation

Hirshfeld surfaces of **Do4OH**, **Do5OH**, **Do6OH I** and **Do6OH II** plotted over *d*_norm_ are shown in Appendix A. The most characteristic feature in all four crystal structures are short O‧‧‧H intermolecular contacts, depicted by red dots in the vicinity of hydroxylic oxygens, belonging to the O-H‧‧‧O hydrogen bonds as the most important interaction between the neighboring molecules. Interestingly, in two representatives of the even series, i.e., in **Do4OH** and **Do6OH I**, there are two symmetry-independent molecules in each crystal structure. In the former, the Hirshfeld surfaces of both molecules are very similar, with one close interaction around hydroxylic oxygen on each end of the **Do4OH** molecules. The same holds true also for **Do5OH**. However, the Hirshfeld surfaces of symmetry-independent molecules in **Do6OH I** do differ: one mimics the close contacts, as are in **Do4OH** and **Do5OH**, but in the other molecule of **Do6OH I**, two close contacts appear at each end of the molecule. The same pattern is present also in the **Do6OH II** polymorph (unluckily, it is impossible to orientate the molecule in a way that would enable both pairs of short contacts to be spotted simultaneously).

The full 2D fingerprint plots are depicted in Appendix A. Despite differences in the packing arrangements of molecules in the studied crystal structures, the 2D fingerprint plots show that the dominant contributions to the Hirshfeld surface are H‧‧‧H, C‧‧‧H/H‧‧‧C and O‧‧‧H/H‧‧‧O interactions in all four crystal structures. These interactions contribute at least 97% to the total Hirshfeld surface. In all four cases, the largest contribution to the total Hirshfeld surface is by H‧‧‧H contacts, from 63.5% in **Do6OH I** to 68.5% in **Do6OH II**. These contacts are graphically represented mostly by turquoise points with the central characteristic wide spike at (*d*_i_ + *d*_e_) ~ 2.2 Å. Two additional characteristic sharp spikes that are close to symmetrical at (*d*_i_ + *d*_e_) ~ 1.65 Å correspond to H‧‧‧O/O‧‧‧H interactions originating either from O-H‧‧‧O or C-H‧‧‧O hydrogen bonds (the former are represented as intense red spots in Appendix A). However, C‧‧‧H/H‧‧‧C interactions that contribute more to the total HS, in comparison with the H‧‧‧O/O‧‧‧H interactions, are more dispersed in 2D fingerprint plots and appear at higher (*d*_i_ + *d*_e_) sums. The percentage contributions of different interactions to the total HS are shown in Figure 6.

The total interaction energies for compounds are separated into different energy contributions—Coulomb, polarization, dispersion and repulsion (Table 3). These partitioned energies are particularly useful to determine the type of intramolecular interactions (for example, *π*∙∙∙*π* stacking is characteristic of large dispersive energy interactions and large values of electrostatic energy can be attributed to hydrogen bonding) [30]. In all four investigated compounds, the largest contribution to the total energy value comes from dispersive interactions, indicating the importance of weak C−H∙∙∙C, C−H∙∙∙*π* and *π*∙∙∙*π* interactions in the overall stability of these compounds. Another important contribution to total energy comes from electrostatic interactions, which can be attributed to O−H∙∙∙O hydrogen bond interactions of terminal hydroxide moieties. Considering the individual contribution of specific energies to total energy, there is a discrepancy in the calculated individual energy in the **Do6OH II** compound in comparison to other compounds. This is especially true for dispersion energy with a value of −129.08 kJ/mol. This could be explained by a weaker contact of molecules in the crystal due to a deviation from planarity, which decreases the number of aromatic interactions. From the percentage contributions of individual energies, it can be concluded that overall stability in planar compounds (**Do4OH** and **Do6OH I**) is balanced between dispersive and electrostatic interactions, while in non-planar (**Do5OH** and **Do6OH II**), the largest contribution to stability comes from dispersion interactions. **The** values of the total calculated energies for **Do6OH I** and **II** are −188.29 kJ mol^−1^ and −128.53 kJ mol^−1^. Lattice energies are estimated to be −208 kJ mol^−1^ and −194 kJ mol^−1^, and −205.2 kJ mol^−1^ and −200.7 kJ mol^−1^ from Pixel calculations. These results show that the more stable form of the **Do6OH** is form **I**, which was also observed in the DSC experiments (vide infra). Considering the lattice energy calculations, it is interesting to note that the energies calculated by Pixel and CrystalExplorer correlate quite well, with the largest difference calculated for **Do5OH**. Similar results with other computational methods and different physical characteristics were previously reported [31]. Unfortunately, the crystal structures of the n = **1** and **3** chain analogs were determined without hydrogen atoms, and the structure of the chain analog n = **2** is disordered; therefore, a detailed analysis of intermolecular interactions was not conducted.

### 3.3. Thermal Analysis (TG/DSC)

The TG curves of the **Do4OH** and **Do5OH** compounds indicate one-step thermal decomposition close to 300 °C with an accompanying exothermic maximum on the DSC curve (black curves on Appendix A). The endothermic events at 98.23 °C (Δ*H_fus_* = 42.6 kJ mol^–1^) and 63.77 °C (Δ*H_fus_* = 37.33 kJ mol^–1^) are attributed to the melting points of **Do4OH** and **Do5OH**, respectively (Appendix A).

On the DSC curve of **Do6OH II** (Figure 7), a distinct endothermic peak is observed at 60 °C, Δ*H_fus_* = 27.09 kJ mol^–1^), which is attributed to the melting point. Two endothermic thermal events are observable on the DSC curve of form **I** (Figure 7). The results of the DSC analysis indicate the presence of both forms in the investigated sample. The first thermal event can be attributed to the melting point of **II**, while the second thermal event is the melting point of **I**. The deconvolution was used to separate these two thermal events and calculate the fusion enthalpy for **I** (Δ*H_fus_* = 17.84 kJ mol^–1^). The calculated melting entropies for **I** and **II** are 0.053 kJ mol^−1^ K^−1^ and 0.081 kJ mol^−1^ K^−1^, respectively. The Gibbs free energy diagram (Appendix A) was constructed over the temperature range of 100 K–400 K. The results indicate that two Gibbs free energy curves intersect at approximately 150 K. The heat of fusion rule, entropy of fusion rule [30] and Gibbs energy diagram indicate the enantiotropic relationship between two polymorphs [31,32,33].

To further explain the odd–even effect in these compounds, a correlation of densities obtained by diffraction experiments, melting points and total energies with a number of methylene groups was carried out (Figure 8, Table 1 and Table 3). Although the data are not complete, some conclusions may be drawn: (i) the melting point of compounds decreases with a larger number of methylene groups; (ii) odd-numbered compounds display systematically lower melting points than even-numbered analogs, (iii) compounds with higher densities display lower melting points, and even-membered compounds of a particular series (i.e., n = **3** and **4**) display lower densities and higher melting points; (iv) the total interaction energy is larger in even-membered analogs; (v) lattice energy calculations do not correlate to melting points. These conclusions are rather difficult to reconcile with the previously mentioned investigations of the odd–even effect. For example, in *α*,*ω*-alkendiols [8], the melting point increases with the number of methylene groups and decreases with lower densities. In terms of the relationship between molecular structure and melting point, it is obvious that planar systems have higher melting points (i.e., n = **3** (non-planar) and **4** (planar)), but at the same time have lower densities, which indicates less-effective crystal packing.

The question is, why do less-effectively (lower density) packed molecules in the crystal structure display higher melting points, when it is usually just the opposite? A plausible explanation of this phenomenon can be found in the percentage contribution of specific contacts calculated by the Hirschfeld analysis (Figure 6). In more densely packed molecules (**Do5OH** and **Do6OH II**), the contribution of close H‧‧‧H contacts is higher, and for O‧‧‧H and C‧‧‧H contacts it is lower. This indicates that close H‧‧‧H contacts act repulsively and reduce the overall stability of the crystal structure. The influence of O‧‧‧H and C‧‧‧H contacts is just the opposite. If correlated with molecular structure, it can be concluded that planar molecules (**Do4OH** and **Do6OH I**) are forming more favorable attractive (O‧‧‧H) contacts, and at the same time diminishing unfavorable repulsive H‧‧‧H contacts. The cost of such interplay is a lower density, but higher melting points. This research also indicates that the generally accepted rule, higher density = higher stability = higher melting point, is not always valid, especially in the case of systems more complex than the simple *α*, *ω*-substituted aliphatic chains. A quite nice example of this deviation from the rule is two polymorphic forms of **Do6OH**–Form **I,** with a higher melting point and lower density than **II**.

### 3.4. Spectroscopy

IR spectra of compounds are given in ESI, Appendix A. The broad OH stretching vibrations are located in the range from 3130 cm^−1^ to 3380 cm^−1^ in all compounds. Vibrations close to 1050 cm^−1^ in all spectra can be attributed to the C–OH group and vibrations close to 1250 cm^−1^ are due to C_aromatic_–O–C_aliphatic_ stretching vibration. Maxima typical for the *ortho*-substituted benzene ring at 750 cm^−1^ and the stretching vibrations of the benzene ring at 1600 cm^−1^ can be observed in the spectrum. Some minor differences are observable in the intensity and position of OH stretching vibration in all compounds and spectra of two polymorphs (Appendix A). These differences can be explained by different hydrogen bond interaction patterns in compounds.

### 3.5. Cyclic Voltammetry

Cyclic voltammetry was used to study the electrochemical properties of the synthesized compounds and to obtain information regarding the charge transfer and structure of the investigated compounds. Earlier studies have shown that odd-numbered molecules impede the charge transfer in *n*-alkenthiolates [10]. Dhiman et al. have also found that the radical scavenging and antioxidant activity of different isomers of hydroxybenzyl alcohols are structure-dependent [34]. In Figure 9, cyclic voltammograms of **Do4OH** (Figure 9A), **Do5OH** (Figure 9B) and **Do6OH** (Figure 9C) are shown. In all voltammograms, one oxidation peak current (A1) at the potential *E*_p,a_ = 0.52 V can be observed, which corresponds to the oxidation of hydroxyl groups in the investigated compounds [35]. This oxidation peak was the most pronounced for the odd-numbered compound **Do5OH**, where the oxidation peak current *I*_pa_ = 30.6 μA, which could be explained by an enhanced charge transfer in this compound. This effect was less visible in even-numbered **Do4OH** and **Do6OH** compounds, where the oxidation peak current was *I*_p,a_ = 24.4 μA and 29.2 μA, respectively.

In Figure 10, the effect of scan rate on the oxidation peak potential and oxidation peak current of all investigated compounds was examined. It was determined that the oxidation peak current and the oxidation peak potential increase with the increase of scan rate for all investigated compounds.

In Figure 11, the logarithms of the anodic peak currents (log *I*_p_,_a_) as a function of the logarithms of the scan rates (log *v*) for the **Do4OH** (Figure 11A), **Do5OH** (Figure 11B) and **Do6OH** (Figure 11C) compounds are shown. The linear correlation was obtained for all four compounds (*R*^2^ = 0.995 for the **Do4OH**, 0.9953 for the **Do5OH** and 0.9923 for the **Do6OH** compound) with a slope around 0.83, which confirmed that the oxidation process in all investigated compounds is under mixed adsorption and diffusion control [36].

### 3.6. DPPH Assay

The antioxidant activities of synthesized compounds (**Do4OH**, **Do5OH** and **Do6OH**) were investigated with a DPPH assay in order to study the correlation of the chain length of the investigated compounds with their antioxidant activity. The obtained results are shown in Figure 12. It can be concluded that only compound **Do5OH** showed small antioxidant activity (% DPPH = 2.4 ± 0.2), while other compounds showed no antioxidant activity. This result agrees with the result of cyclic voltammetry since the odd-numbered compound **Do5OH**, which showed the most-pronounced oxidation peak due to the enhanced charge transfer, showed antioxidant activity, while the even-numbered Do4OH and Do6OH compounds showed no antioxidant activity.

## 4. Conclusions

Three novel aromatic alcohol chain isomers were prepared and structurally characterized. In all compounds, two benzyl alcohol units are connected by an aliphatic chain of different lengths, ((CH_2_)_n_; n = **4**, **5** and **6**). The major differences in the molecular structures were found in the overall planarity of the molecules and in the conformation of the aliphatic chain. Molecules with an even number of CH_2_ groups are planar with the all-*trans* conformation of the aliphatic chain, and odd-numbered molecules are non-planar, with partial *gauche* conformation (with the exception of **Do6OH II**). As a consequence of these structural modifications, odd-numbered compounds display systematically lower melting points. The comparison with previously reported chain analogs (n = **1**, **2** and **3**) indicates a decrease in melting point with an increase in the number of CH_2_ groups, but with higher melting points of particular even-numbered molecules. Interestingly, higher melting point compounds of particular odd–even pairs display lower density, which is in contradiction to previously studied cases of the odd–even effect. It was found that more densely packed molecules (even-membered) show a higher percentage of repulsive H‧‧‧H contacts, which is a plausible explanation for the lowering of the melting point. The electrochemical study has shown one oxidation peak which corresponds to the oxidation of hydroxyl groups in all compounds. The highest oxidation peak current and moderate antioxidant activity were found for the non-planar compound (**Do5OH**), indicating the importance of the conformational properties of molecules in antioxidant activity.

## Figures and Tables

**Figure 1 molecules-27-03781-f001:**
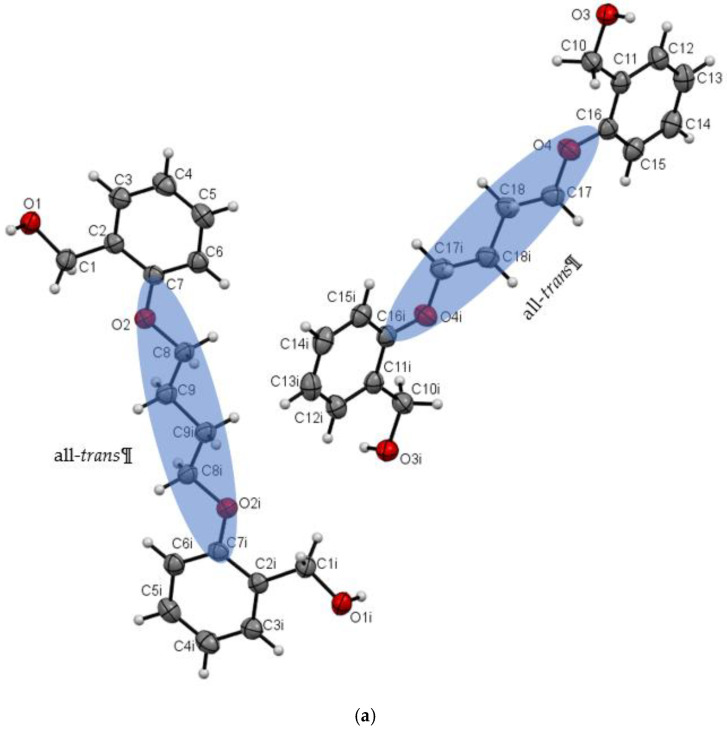
(**a**) ORTEP plot of **Do4OH** with displacement ellipsoids of non-hydrogen atoms drawn at the 50% probability level. The conformation of the aliphatic chain is highlighted by the transparent blue ellipse. (**b**) ORTEP plot of **Do5OH** with displacement ellipsoids of non-hydrogen atoms drawn at the 50% probability level. Representation of Newman projections looking down the O2–C8, C8–C9 and C9–C10 bonds indicating a partial gauche conformation of the aliphatic chain in Do5OH. I = -x, -y, -z.

**Figure 2 molecules-27-03781-f002:**
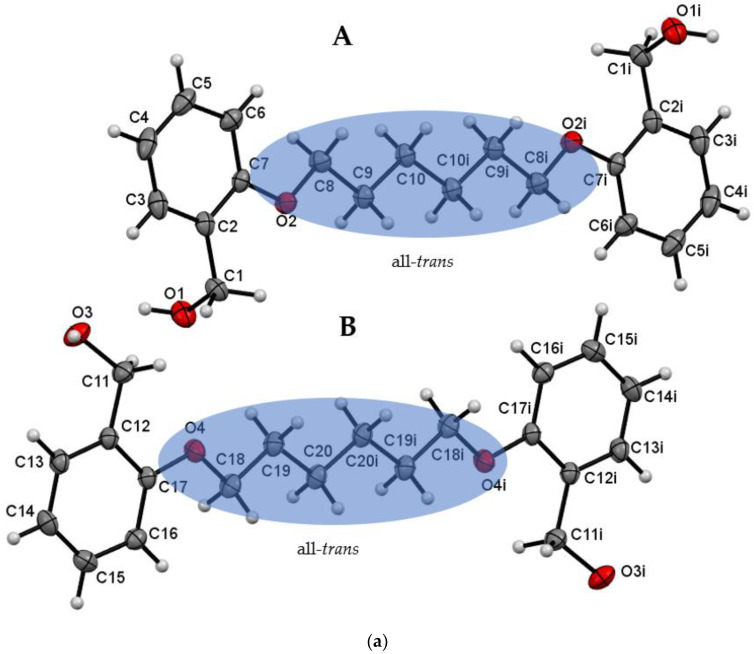
(**a**) ORTEP plot of **Do6OH I** with displacement ellipsoids of non-hydrogen atoms drawn at the 50% probability level (i = -x, -y, -z). The conformation of the aliphatic chain is highlighted by the transparent blue ellipse. (**b**) ORTEP plot of **Do6OH II** with displacement ellipsoids of non-hydrogen atoms drawn at the 50% probability level. The representation of the Newman projections of O2–C8, C8–C9 and C9–C10 bonds indicating a partial gauche conformation of the aliphatic chain in **Do6OH II** (i = -x, -y, -z).

**Figure 3 molecules-27-03781-f003:**
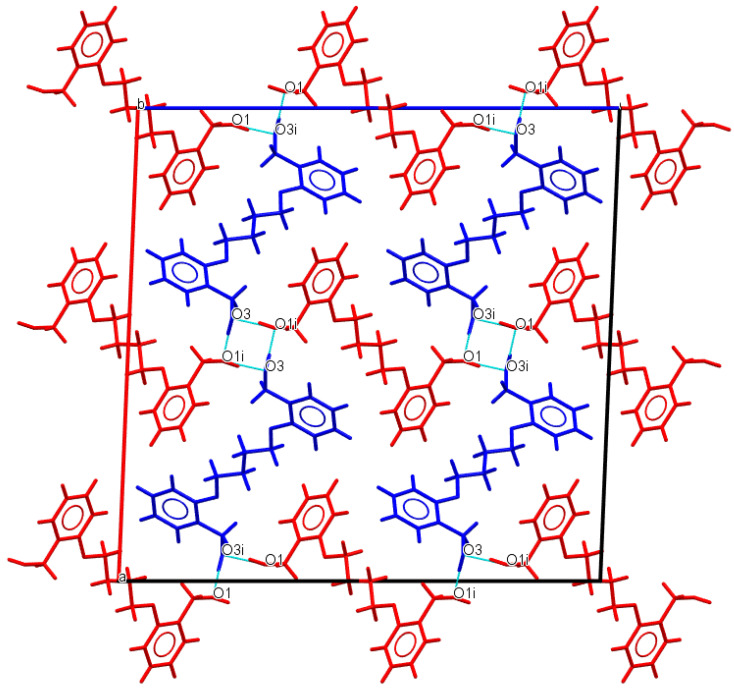
Representation of O−H∙∙∙O hydrogen bond interactions in **Do4OH** (blue dashed lines) along *b*-axis. Two symmetrically independent molecules are represented by blue and red colors.

**Figure 4 molecules-27-03781-f004:**
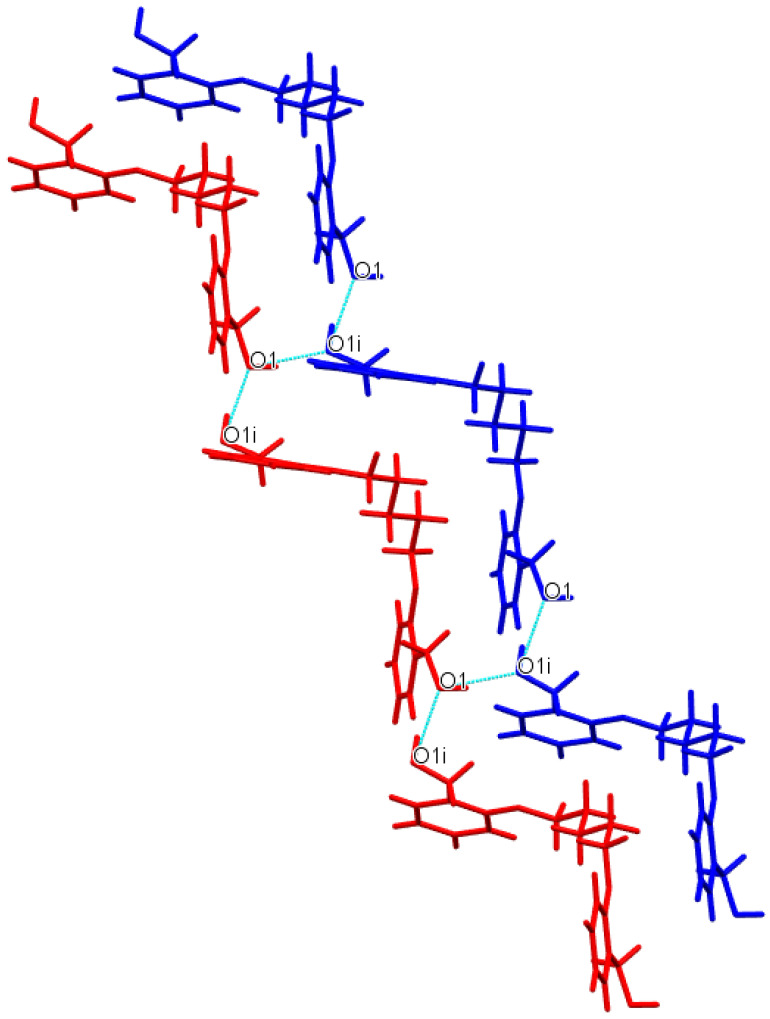
Representation of staircase-like motif in **Do5OH**. O−H∙∙∙O hydrogen bond interactions are represented by blue dashed lines and adjacent chains by blue and red colors.

**Figure 5 molecules-27-03781-f005:**
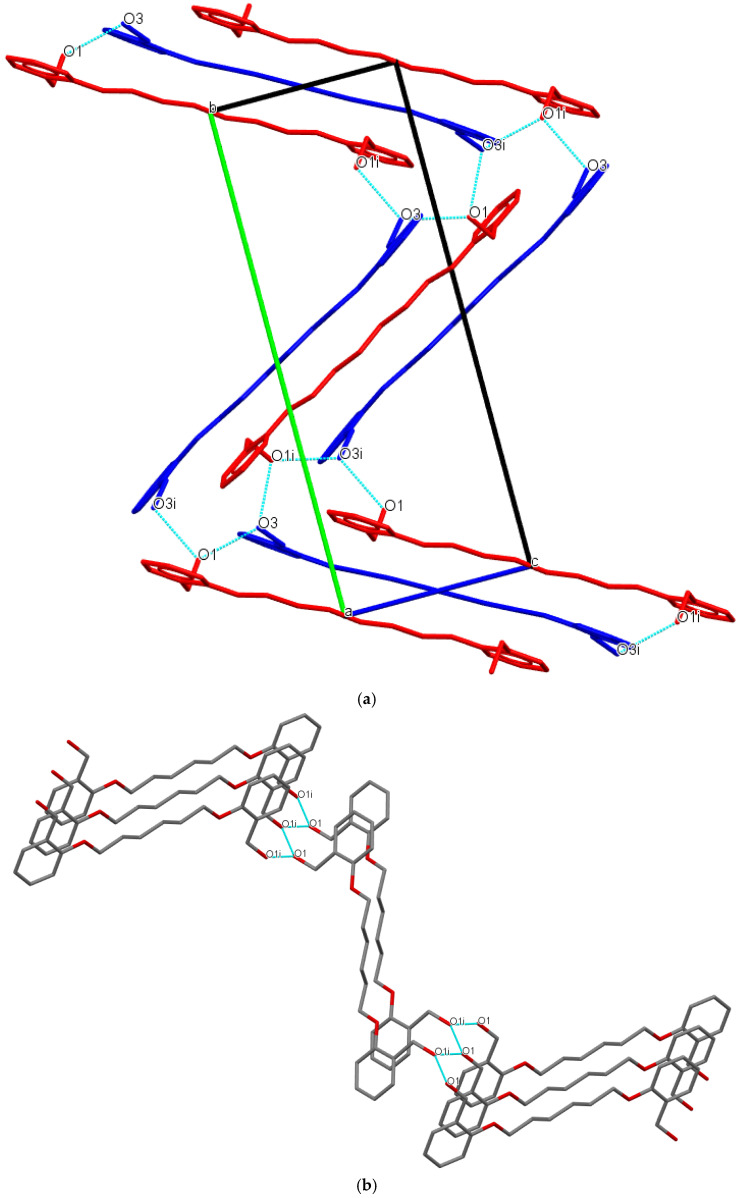
(**a**) View of the staircase packing arrangement in **Do6OH I**. O−H∙∙∙O hydrogen bond interactions is represented by blue dashed lines. Conformer molecules are red (**A**) and blue (**B**) colored and hydrogen atoms are omitted due to clarity. (**b**) Representation of 2D sheet motif in **Do6OH II**. O−H∙∙∙O hydrogen bond interactions are represented by blue dashed lines and hydrogen atoms are omitted due to clarity.

**Figure 6 molecules-27-03781-f006:**
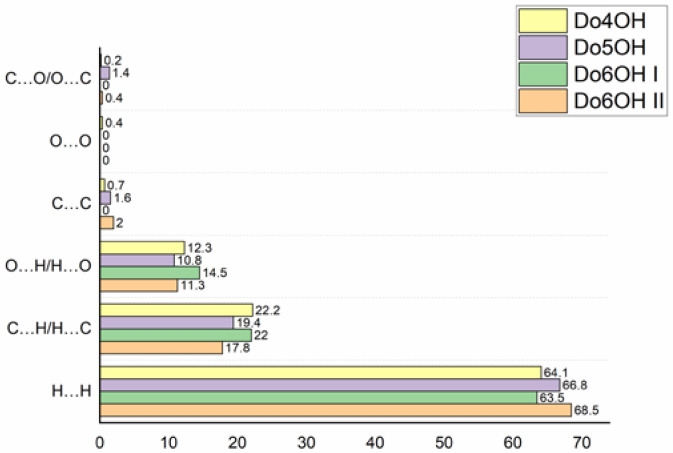
The percentage contributions of different interactions to the total Hirshfeld surface in four studied compounds.

**Figure 7 molecules-27-03781-f007:**
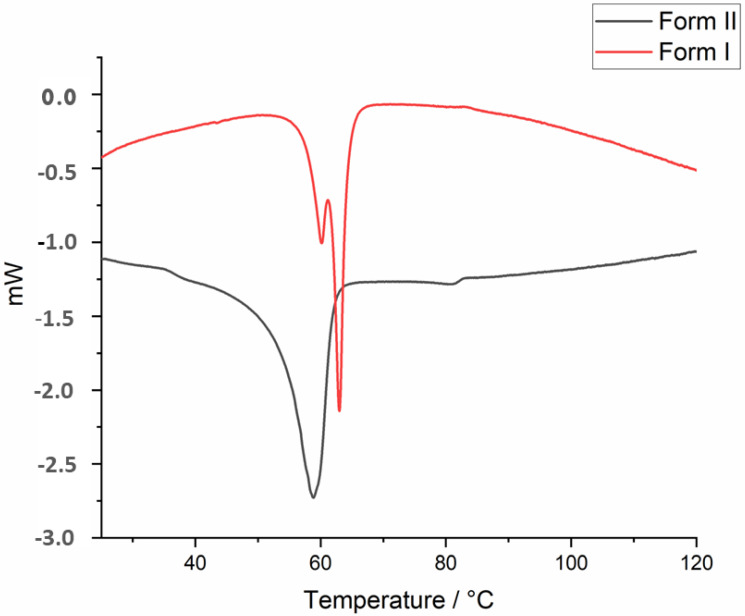
DSC curves of **Do6OH** forms **I** and **II**.

**Figure 8 molecules-27-03781-f008:**
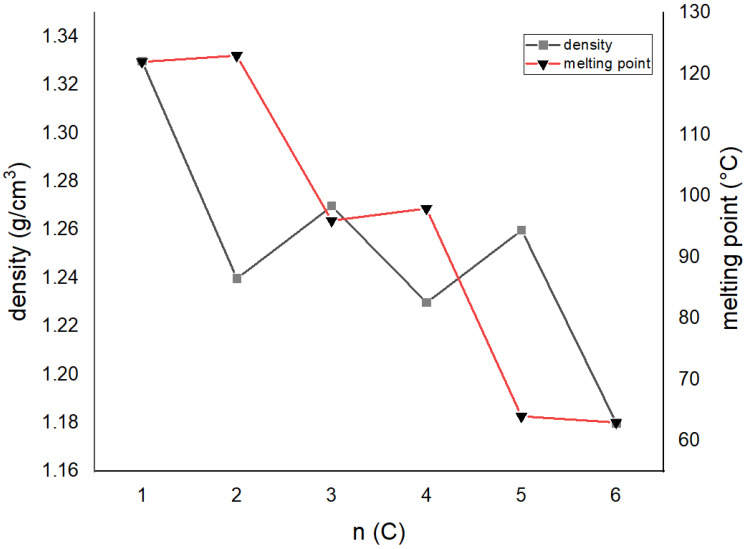
Melting point and density alteration with an increasing number of methylene groups in the aliphatic chain (only data for **Do6OH I** are presented).

**Figure 9 molecules-27-03781-f009:**
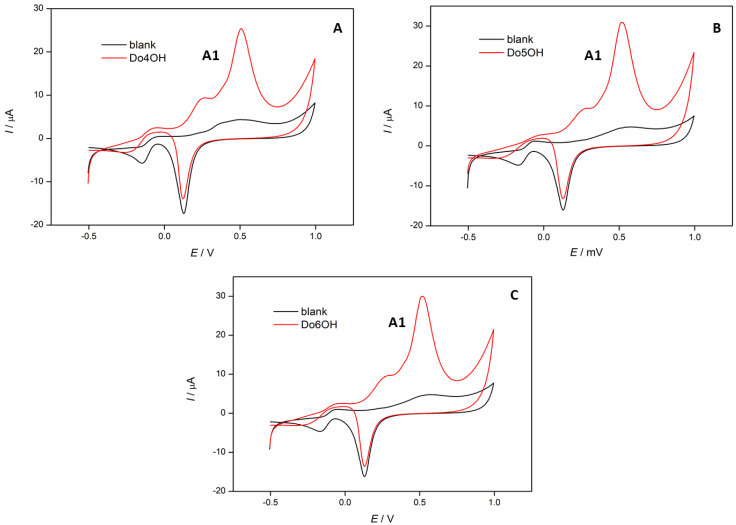
Cyclic voltammograms of investigated compounds (*c* = 1 × 10^−4^ M): (**A**) **Do4OH**, (**B**) **Do5OH** and (**C**) **Do6OH**, recorded in 0.1 M NaOH. Scan rate 100 mV/s.

**Figure 10 molecules-27-03781-f010:**
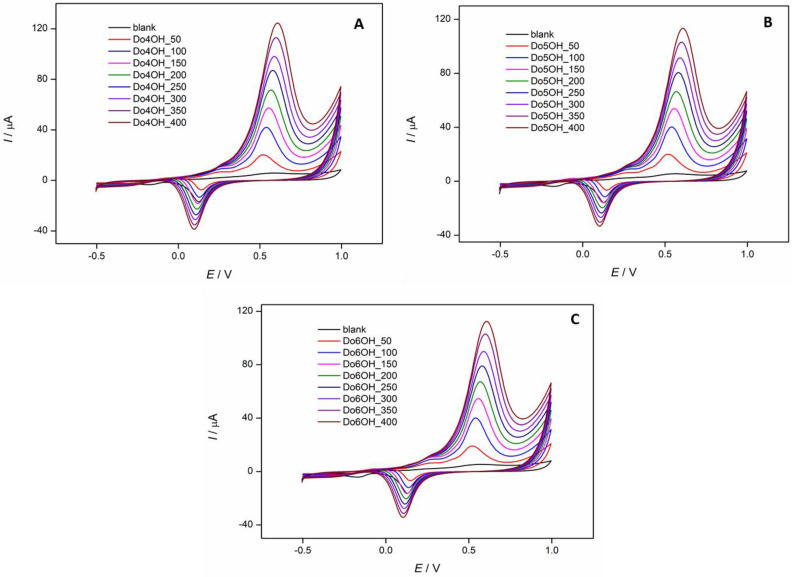
Cyclic voltammograms of investigated compounds (*c* = 1 × 10^−4^ M): (**A**) **Do4OH**, (**B**) **Do5OH** and (**C**) **Do6OH**, recorded in 0.1 M NaOH at different scan rates (*v* = 50–400 mV/s).

**Figure 11 molecules-27-03781-f011:**
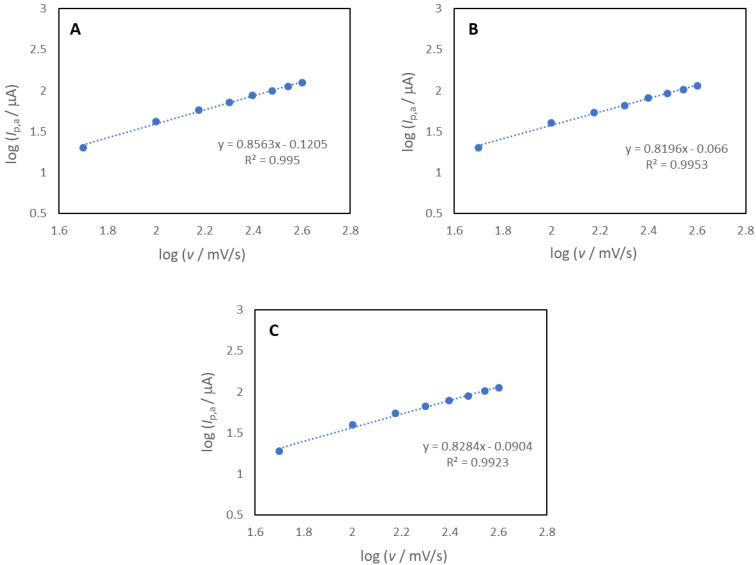
The logarithm of peak current (log *I*_p_) as a function of the logarithm of scan rate (log *v*) for the (**A**) **Do4OH**, (**B**) **Do5OH** and (**C**) **Do6OH** compounds.

**Figure 12 molecules-27-03781-f012:**
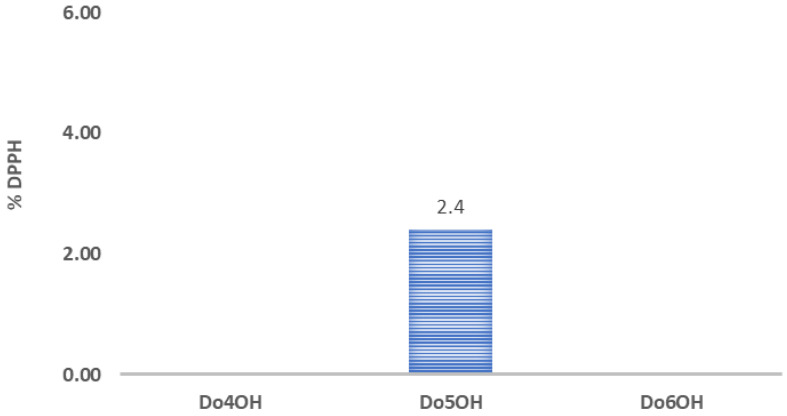
Column graphs showing % DPPH inhibition of all investigated compounds.

**Table 1 molecules-27-03781-t001:** Selected parameters for investigated compounds.

Compound	The Dihedral Angle between Benzene Rings/°	Aliphatic Chain Conformation	OH‧‧‧OH Distance /Å	Density/g cm^−3^	Melting Point/°C	CSD Code
1,2-bis[2-(hydroxyformyl)phenoxy]methylene	59.43	*gauche*	9.815(2)	1.332	121–123 ref. [26]	FASCUY [27]
1,2-bis[2-(hydroxyformyl)phenoxy]ethylene	60.89	*trans–gauche*–*trans*	6.446(4)	1.242	122–124 ref. [23]	SAZCIF [28]
1,2-bis[2-(hydroxyformyl)phenoxy]propylene	80.73 84.59	*trans*–*gauche*-*gauche*–*trans*	9.659(2) 9.725(2)	1.274	96 ref. [23]	FASDAF [27]
**Do4OH**	0	all-*trans*	13.754(2)	1.236	98	This work
**Do5OH**	84.2(1)	*trans*–*gauche*–*trans*–*gauche*–*trans*	11.133(2)	1.268	64	This work
**Do6OH I**	0	all-*trans*	13.954(2)–conformer A 15.799(2)–conformer B	1.188	63	This work
**Do6OH II**	0	*trans*–*gauche*–*trans*–*trans*–*trans*–*gauche*-*trans*	13.461(2)	1.242	60	This work

**Table 2 molecules-27-03781-t002:** Hydrogen bond geometry (Å, °) for **Do4OH, Do5OH, Do6OH I** and **Do6OH II**.

D−H∙∙∙A	*d*(D—H)	*d*(H⋅⋅⋅A)	*d*(D⋅⋅⋅A)	∠(D—H⋅⋅⋅A)	Symmetry code
**Do4OH**					
O1—H1···O3	0.976(3)	1.678(3)	2.648(3)	172(2)	-x+1,+y-1/2,-z+1/2
O3—H3A···O1	0.946(3)	1.708(3)	2.653(2)	176(3)	x,+y+1,+z
**Do5OH**					
O1—H1···O1	0.883(2)	1.850(2)	2.710(1)	164(2)	-x+1/2,+y-1/2,+z
**Do6OH I**					
O3—H3A···O1	0.875(2)	1.802(2)	2.669(1)	170(2)	x,+y,+z-1
O1—H1···O3	0.884(2)	1.813(2)	2.687(1)	169(2)	x,-y+1/2,+z+1/2
**Do6OH II**					
O3—H1···O1	0.880(2)	1.790(2)	2.658(1)	168(2)	x-1/2,+y,-z+1/2+1

**Table 3 molecules-27-03781-t003:** Molecular pair interaction total energy (E_tot_) and energies partitioned into electrostatic (E_ele_), polarization (E_pol_), dispersion (E_dis_) and repulsion (E_rep_) contributions (kJ/mol) for individual compounds. Percentage contributions of individual attractive interaction energies to total energies are given in brackets.

Compound	E_ele_	E_pol_	E_dis_	E_rep_	E_tot_	E_latt_	PixelC
**Do4OH**	−151.99 (44.5%)	−24.79 (7.3%)	−164.97 (48.2%)	148.81	−192.94	−192	−188.7
**Do5OH**	−80.01 (31.5%)	−13.39 (5.3%)	−161.39 (63.2%)	97.39	−157.40	−193	−215.2
**Do6OH I**	−150.09 (44.9%)	−23.45 (7.2%)	−160.96 (47.9%)	146.22	−188.29	−208	−205.2
**Do6OH II**	−77.48 (35.3%)	−12.13 (5.2%)	−129.08 (59.5%)	90.16	−128.53	−194	−200.7

## Data Availability

Not aplicable.

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
