# Peer review of "Influence of Aliphatic Chain Length on Structural, Thermal and Electrochemical Properties of n-alkylene Benzyl Alcohols: A Study of the Odd–Even Effect"

_molecules, 2022, doi:10.3390/molecules27123781_

Round 1
Reviewer 1 Report
The article “Influence of aliphatic chain length on structural, thermal and electrochemical properties of 1,2-bis[2-(hydroxymethyl)phenoxy] n-alkylene aromatic alcohols: study of odd-even effect ”, authors Tomislav Balić, Marija Paurević, Marta Počkaj, Martina Medvidović-Kosanović , Dominik Goman, Aleksandar Széchenyi, Zsolt Preisz and Sándor Kunsági-Máté, reports the synthesis and the characterization aspects of three novel aromatic alcohol with the two benzyl alcohol units connected by an aliphatic chain of different lengths.
The odd-even effect was expected to be highlighted through theoretical and experimental methods. Different aspects were followed in the paper: the planarity of the systems, the crystal packing, the melting points, the density of the compounds, total interaction and lattice energies. The authors ascertained the violation of the well-known rule which states that a higher density means higher stability and also a higher melting point. This discrepancy was explained through the percentage contributions of H-H, O-H and C-H close contacts.
The following mentions must be taken into account :
-line 115: the terms I and II must be explained, so the lines 229-230 are more suitable in the 2.1 section
-line 149: the reference [20] must be mentioned
- line 161: Even though the reference was mentioned, a brief description of the procedure followed in this paper is recommended
- in Table 2, the first two values of d(D-A) are higher than the ones found in the scientific literature. There should be more clarifications regarding why the authors decided to consider them as H bounds.
-line 356: Fig 8 is not really necessary ( S4 and S5 describe the DSC curves of the two compounds)
- line 375+ Figure 10: What is the reason for not calculating the Total energy for all the compounds? There were meant to be some theoretical calculations.
- lines 379-380: “even-membered compounds…(n=3)”
Generally speaking, the values from the tables should be explained more and correlated with the behavior of the compounds. A clear example of this is Table 3. Additional information needs to be added, especially in the case of the dispersion energy for the Do6OH II, where the value is drastically different, with no further explanation given.
In the Electronic Supplementary Information it is recommended to resume the meaning of the table parameters, even though those are already mentioned in the article's body.
- S3: It is not mandatory to specify the temperature both in °C and in K.
- S4: the 4th column should be deleted because the article already refers to the functional used, namely B3LYP
Author Response
Answer to reviewers` comments
The authors thank the reviewer for their insightful comments Changes made in the manuscript are marked in red color.
Answers to reviewer 1
- line 115: the terms I and II must be explained, so the lines 229-230 are more suitable in the 2.1 section
This is corrected.
- line 149: the reference [20] must be mentioned
This is corrected, the reference:
- J. Turner; J. J. McKinnon; S. K. Wolff; D. J. Grimwood; P. R. Spackman; D. Jayatilaka; M. A. Spackman CrystalExplorer17; University of Western Australia, 2017.
is mentioned in line 129.
- line 161: Even though the reference was mentioned, a brief description of the procedure followed in this paper is recommended
A brief description of the calculation procedure is added to the manuscript. The following references were added:
[20] Neese, F. (2018). WIREs Comput. Mol. Sci. 8, e1327
[21] Lu, T. & Chen, F. (2012). J. Comput. Chem. 33, 580–592.
- In Table 2, the first two values of d(D-A) are higher than the ones found in the scientific literature. There should be more clarifications regarding why the authors decided to consider them as H bounds.
Hydrogen bonds (weak interactions) with large D-A values were removed from Table 2. Criteria for D-A distance for hydrogen bonds is from 2.2 to 3.5 Å. Bonds outside of this distance (3-4 Å) are considered to be weak Van der Waals interactions. This is mentioned and corrected in the manuscript.
- line 356: Fig 8 is not really necessary ( S4 and S5 describe the DSC curves of the two compounds)
This is corrected as requested.
- line 375+ Figure 10: What is the reason for not calculating the Total energy for all the compounds? There were meant to be some theoretical calculations.
The crystal structures of two previously published compounds did not contain hydrogen atoms, and therefore we haven’t calculated lattice energy and total energy. The calculations can be performed but the data obtained in such a manner are not reliable.
- lines 379-380: “even-membered compounds…(n=3)”
The part in the bracket is related to the expression “particular series”, not even-membered.
- 8. Generally speaking, the values from the tables should be explained more and correlated with the behavior of the compounds. A clear example of this is Table 3. Additional information needs to be added, especially in the case of the dispersion energy for the Do6OH II, where the value is drastically different, with no further explanation given.
We have considered this and added a comment in the main article:
„Considering the individual contribution of specific energies to total energy, there is a discrepancy in calculated individual energy in Do6OH II compound in comparison to other compounds. This is especially true for dispersion energy displaying a value of –129.08 kJ/mol. This could be explained by weaker contact of molecules in the crystal due to deviation from planarity, which decreases the number of aromatic interactions. From the percentage contribution of individual energies, it can be concluded that overall stability in planar com-pounds (Do4OH and Do6OH I) is balanced between dispersive and electrostatic interactions, while in non-planar (Do5OH and Do6OH II) the largest contribution to stability comes from dispersion interactions.“
In Table 3, the percentage contribution of individual attractive interaction energies to total energies are given in brackets for Do4OH, Do5OH, and Do6OH.
- In the Electronic Supplementary Information it is recommended to resume the meaning of the table parameters, even though those are already mentioned in the article's body.
This is corrected as requested.
- S3: It is not mandatory to specify the temperature both in °C and in K.
This is corrected, only temperature in K is presented in Table S3.
- S4: the 4th column should be deleted because the article already refers to the functional used, namely B3LYP.
This is corrected, the 4th column is deleted in table S4.

Reviewer 2 Report
Before the Editor makes a decision, I suggest that the authors must take into account the following corrections:
- I think the title needs to be reformulated to become more “friendly”. Maybe the "1,2-bis[2-(hydroxymethyl)phenoxy] n-alkylene" expression is not necessarily needed
- The "Abstract" section should be more concise.
- It is not clear how were obtained the numerical data used in Section 2.
- From where were taken the data used in Tables and graphic representations?
- Authors must try to reduce the number of figures.
- Some editing "glitches" need to be corrected.
- I think the authors need to emphasize more clearly the contribution of the manuscript from a scientific point of view.
- References are carelessly edited.
- Also, references are not uniformly written. In some references the name of the journal is incorrect abbreviated.
- I think, the author must strengthen the References section with some articles that use some similar techniques, to make the techniques used more plausible, for instance: A novel model of plane waves of two-temperature fiber-reinforced thermoelastic medium under the effect of gravity with three-phase-lag model, Int J Numer Method H, 29(12), 4788-4806, 2019; Recent trends in computational fluid dynamics, Front Phys, 8 (2020), 3389/fphy.2020.593111
If the authors take into account all these corrections, then this manuscript deserves to be published.
Author Response
Answer to reviewers` comments
The authors thank the reviewer for their insightful comments. Changes made in the manuscript are marked in red color.
Answers to reviewer 2
- I think the title needs to be reformulated to become more “friendly”. Maybe the "1,2-bis[2-(hydroxymethyl)phenoxy] n-alkylene" expression is not necessarily needed
The title is corrected as suggested by the reviewer.
- The "Abstract" section should be more concise.
The Abstract is presented in a more informative and concise form.
- It is not clear how were obtained the numerical data used in Section 2.
The authors thank the reviewer for this question. Unfortunately, we do not fully understand this comment. The compounds were prepared as described in the manuscript and all analytical data were obtained as described in the manuscript (Experimental section). Calculations of interaction energies were done as described (Crystal Explorer and Oscail – PixelC). Data for previously published structures were obtained via the CCDC database as cif files.
- From where were taken the data used in Tables and graphic representations?
The data used for article preparation were obtained by single-crystal diffraction experiments. The final structures were deposited in the CCDC database under 2161717–2161720 numbers. Graphic representations were done using Mercury and calculations presented in the article using Mercury and PLATON programs. Data from previously published structures were obtained as cif files from the CCDC database. The relevant literature is cited in the article.
- Authors must try to reduce the number of figures.
This is corrected. Fig. 6 is moved to ESI (Fig. S4) and Fig 8. is removed from the manuscript as proposed by reviewers.
- Some editing "glitches" need to be corrected.
The article is corrected for mistakes.
- I think the authors need to emphasize more clearly the contribution of the manuscript from a scientific point of view.
We have corrected the abstract and conclusions in order to emphasize our findings and scientific contribution.
- References are carelessly edited.
It is corrected.
- Also, references are not uniformly written. In some references, the name of the journal is incorrect abbreviated.
It is corrected. Abbreviated journal titles have been corrected according to WOS Journal Title Abbreviations. DOI numbers are deleted, also some names are corrected.
- I think, the author must strengthen the References section with some articles that use some similar techniques, to make the techniques used more plausible, for instance: A novel model of plane waves of two-temperature fiber-reinforced thermoelastic medium under the effect of gravity with three-phase-lag model, Int J Numer Method H, 29(12), 4788-4806, 2019; Recent trends in computational fluid dynamics, Front Phys, 8 (2020), 3389/fphy.2020.593111
It is corrected and a suggested reference is added
Reviewer 3 Report
Balic et al. prepared a series of 4 new aromatic alcohols and systematically compared their single crystal structures in relation to emerging properties such as melting point, density, and energy. Conclusions are drawn and associated with the odd/even effect.
In my opinion, the experiments and analysis of the results are impeccable, especially the crystal structures which constitute the essential part of the publication. However, the way the results are presented and discussed cannot be accepted in the current form of the manuscript. First and foremost, this concerns the length of the manuscript. The abstract is excessive for the amount of the findings presented here. The same applies to the introduction and the conclusions of the manuscript (for instance synthetic methods for the production of alcohols are simply not relevant here). Furthermore, details and data are displayed which are simply not expedient to the scope of the work - Cyclic voltammetry and DPPH assays do not have any relation to the odd-even effect, they seem completely detached from the rest of the discussion. If there is a relation, the authors should make that clear. Lastly, some essential passages were hard to understand. The discussion of the conformation is a somewhat hard to follow with the current graphics. I suggest highlighting and modifying the pictures of the crystal structures. The authors should rewrite manuscript to make it succinct and clear and focus on what is claimed in the title.
Author Response
Answer to reviewers` comments
The authors thank the reviewers for their insightful comments. Changes made in the manuscript are marked in red color.
Answers to reviewer 3
- The abstract is excessive for the amount of the findings presented here. The same applies to the introduction and the conclusions of the manuscript (for instance synthetic methods for the production of alcohols are simply not relevant here).
We have considered the length of the manuscript and corrected it according to the comments. The abstract, introduction, and conclusion are presented in a more informative and concise form. Some pictures are transferred to ESI and Fig. 8 is removed. The synthetic procedure is transferred to ESI.
- Furthermore, details and data are displayed which are simply not expedient to the scope of the work - Cyclic voltammetry and DPPH assays do not have any relation to the odd-even effect, they seem completely detached from the rest of the discussion. If there is a relation, the authors should make that clear.
The authors thank the reviewer for this question. Cyclic voltammetry and DPPH assay were used to study the structure dependence of electrochemical and antioxidant properties of investigated compounds. Cyclic voltammetry was used to study the electrochemical properties of the synthesized compounds and to obtain information regarding charge transfer and structure of the investigated compounds. Earlier studies by Jiang et al have shown that odd-number molecules impede charge transfer in n-alkenthiolates. Dhiman et al have also found that radical scavenging and antioxidant activity of different isomers of hydroxy benzyl alcohols are structure-dependent. The oxidation peak visible in cyclic voltammograms of investigated compounds was the most pronounced for the odd-number compound Do5OH with non-planar, partial gauche conformation, which could be explained by enhanced charge transfer in this compound. This effect was less visible in even-numbered Do4OH and Do6OH com-pounds due to their planar, all-trans configuration of the aliphatic chain.
Antioxidant activities of synthesized compounds (Do4OH, Do5OH, and Do6OH) were investigated with a DPPH assay to confirm that the antioxidant activity of investigated compounds is structure-dependent. Earlier studies by Dhiman et al have shown that hydroxy benzyl alcohols have antioxidant activity and that it varies depending on the structure of different isomers of hydroxy benzyl alcohols as was obtained in our study.
- Lastly, some essential passages were hard to understand. The discussion of the conformation is a somewhat hard to follow with the current graphics. I suggest highlighting and modifying the pictures of the crystal structures. The authors should rewrite manuscript to make it succinct and clear and focus on what is claimed in the title.
We have modified Figs. 1 and 2 in order to highlight the conformation of compounds. The manuscript is rewritten to make it more clear.
Reviewer 4 Report
In my opinion all structural issues are described in a very detailed way and indicate interesting trends within the groups of compounds that contain longer aliphatic chains. The motivation for the CV studies and DPPH assay needs improvement. The authors should quote relevant papers about the antioxidant activity of benzyl alcohols and compare their results with the published ones. Why is benzyl alcohol considered an antioxidant and their compounds do not exhibit similar properties bearing two similar functional groups? Please, try to improve the discussion on the topic.
Author Response
Answer to reviewers` comments
The authors thank the reviewer for their insightful comments. Changes made in the manuscript are marked in red color.
Answers to reviewer 4
- In my opinion all structural issues are described in a very detailed way and indicate interesting trends within the groups of compounds that contain longer aliphatic chains. The motivation for the CV studies and DPPH assay needs improvement. The authors should quote relevant papers about the antioxidant activity of benzyl alcohols and compare their results with the published ones. Why is benzyl alcohol considered an antioxidant and their compounds do not exhibit similar properties bearing two similar functional groups? Please, try to improve the discussion on the topic.
The authors thank the reviewer for this question. Cyclic voltammetry and DPPH assay were used to study the structure dependence of electrochemical and antioxidant properties of investigated compounds. Cyclic voltammetry was used to study the electrochemical properties of the synthesized compounds and to obtain information regarding charge transfer and structure of the investigated compounds. Earlier studies by Jiang et al have shown that odd-number molecules impede charge transfer in n-alkenthiolates. Dhiman et al have also found that radical scavenging and antioxidant activity of different isomers of hydroxy benzyl alcohols are structure-dependent. The oxidation peak visible in cyclic voltammograms of investigated compounds was the most pronounced for the odd-number compound Do5OH with non-planar, partial gauche conformation. It could be explained by enhanced charge transfer in this compound. This effect was less visible in even-numbered Do4OH and Do6OH compounds due to their planar, all-trans configuration of the aliphatic chain.
Antioxidant activities of synthesized compounds (Do4OH, Do5OH, and Do6OH) were investigated with a DPPH assay to confirm that the antioxidant activity of investigated compounds is structure-dependent. Earlier studies by Dhiman et al have shown that hydroxy benzyl alcohols have antioxidant activity and that it varies depending on the structure of different isomers of hydroxy benzyl alcohols as was obtained in our study.
Round 2
Reviewer 1 Report
There is one issue: "van der Waals" is the right spelling and not "Van der Waals"
Author Response
The authors thank all reviewers for their insightful comments. Changes made in the manuscript are marked in red color and yellow highlighted.
Answers to reviewer 1
1.There is one issue: "van der Waals" is the right spelling and not "Van der Waals"
This is corrected
Reviewer 2 Report
The authors considered all my proposed corrections, which led to an improved form of the manuscript.
Author Response
Answers to reviewer 2
- The authors considered all my proposed corrections, which led to an improved form of the manuscript.
Thank you.
Reviewer 3 Report
Balic et al. have tackled the raised points and significantly improved their manuscript. After reading the new version of the manuscript, the following points remain to be addressed:
Page 1, Line 27: It should be “all-trans CONFORMATION”, not configuration. This mistake has been made several times throughout the manuscript.
Page 4, Line 159: Details on how the CSD search was performed should be added.
Page 5, Figure 2. The authors have highlighted the area of interest in the molecule. However, my comment was referring to the orientation of the molecule. The gauche conformations in the respective molecules are hard to recognize in the current orientation. Another representation should be added, where the molecule can be seen from the side (for instance Newman projection).
Page 6, Line 180: The asymmetric unit is composed of two symmetrically independent HALF molecules. The current phrasing is incorrect.
Page 8, Line 200: Space group symbols should be in standard format (without space).
Page 16, Figure 8: All the displayed values actually have units. They should be given here. These are not arbitrary units. Furthermore, only for the chain length 4, 5, and 6 the total energy has been calculated. However, since the crystal structures of 1-3 are available, the total energy should also be calculated for these. Three points in the diagram have little value.
Page 17, Line 393: I fail to see the relevance of the CV measurements. The authors have justified why they did these. Yet, they appear very similar, almost identical. If there are really significant differences, actual values should be given. Furthermore, conclusion about the influence of the conformation (Line 403) can simply not be made. The conformation observed in the solid, is by no means the one present in solution. With this analysis, because it is done in solution, only conclusions about the influence of the chain length can be made and nothing else.
Page 19, Line 426: The same is true for the DPPH assays. There is no information about any isomers in solution, only the conformation in the solid state is known. Furthermore, how are these investigations related to the odd-even effect? This should be explained.
The grammar must be improved. This concerns the usage of tense, wording, and grammatical errors. All of this should be checked again carefully for the whole manuscript.
Author Response
The authors thank the reviewer for their insightful comments. Changes made in the manuscript are marked in red color and yellow highlighted.
Answers to reviewer 3
Balic et al. have tackled the raised points and significantly improved their manuscript. After reading the new version of the manuscript, the following points remain to be addressed:
- Page 1, Line 27: It should be “all-trans CONFORMATION”, not configuration. This mistake has been made several times throughout the manuscript.
This is corrected
- Page 4, Line 159: Details on how the CSD search was performed should be added.
There is a description of database search on page 12 line 254: “To further correlate the properties of investigated compounds, a Cambridge Structural Database (CSD) [29] search was performed for compounds containing 2-(hydroxyformyl)phenoxy units linked by a different number of methylene groups in the aliphatic chain.“
- Page 5, Figure 2. The authors have highlighted the area of interest in the molecule. However, my comment was referring to the orientation of the molecule. The gauche conformations in the respective molecules are hard to recognize in the current orientation. Another representation should be added, where the molecule can be seen from the side (for instance Newman projection).
Our apologies, we did not understand well this point in the previous review. Newman projections of particular bonds of interest are added as part of Figs. 1b. and 2b. Projections were drawn using PLATON.
- Page 6, Line 180: The asymmetric unit is composed of two symmetrically independent HALF molecules. The current phrasing is incorrect.
This is corrected
- Page 8, Line 200: Space group symbols should be in standard format (without space).
This is corrected
- Page 16, Figure 8: All the displayed values actually have units. They should be given here. These are not arbitrary units. Furthermore, only for the chain length 4, 5, and 6 the total energy has been calculated. However, since the crystal structures of 1-3 are available, the total energy should also be calculated for these. Three points in the diagram have little value.
Figure 8 is corrected according to the comment: The total energy curve for 3 compounds is removed from the figure and units for density and melting point are added. Considering calculation of total energy: The crystal structures of two previously published compounds did not contain hydrogen atoms, and therefore we haven’t calculated lattice energy and total energy. The calculations can be performed but the data obtained in such a manner are not reliable. We have added a short comment regarding calculations in the main article. We agree that this might be valuable data but unfortunately, there is no possibility of reliable calculation. One of the reviewers already commented on this previously.
- Page 17, Line 393: I fail to see the relevance of the CV measurements. The authors have justified why they did these. Yet, they appear very similar, almost identical. If there are really significant differences, actual values should be given. Furthermore, conclusion about the influence of the conformation (Line 403) can simply not be made. The conformation observed in the solid, is by no means the one present in solution. With this analysis, because it is done in solution, only conclusions about the influence of the chain length can be made and nothing else.
The authors thank the reviewer for this comment. Our intention was to characterize the investigated compounds with cyclic voltammetry and to check possible differences between the compounds since similar studies were not found. Differences in oxidation peak currents were observed and the actual values are added in the manuscript. Conclusions are rewritten, the influence of the conformation was erased and only the influence of chain length is discussed.
- Page 19, Line 426: The same is true for the DPPH assays. There is no information about any isomers in solution, only the conformation in the solid state is known. Furthermore, how are these investigations related to the odd-even effect? This should be explained.
DPPH assay was performed in order to study the correlation of the chain length of investigated compounds with their antioxidant activity. The obtained result agrees with the result of cyclic voltammetry since the odd-number compound Do5OH, which showed the most pronounced oxidation peak due to the enhanced charge transfer, showed antioxidant activity while even-numbered Do4OH and Do6OH compounds showed no antioxidant activity. This explanation is added to the manuscript.
9.The grammar must be improved. This concerns the usage of tense, wording, and grammatical errors. All of this should be checked again carefully for the whole manuscript.
The English language is corrected throughout the manuscript.